CmHem, a hemolin-like gene identified from Cnaphalocrocis medinalis, involved in metamorphosis and baculovirus infection

Han Guangjie
Li Chuanming
Zhang Nan
Liu Qin
Huang Lixin
Xia Yang
Xu Jian bio-xj@163.com
Lixiahe District Institute of Agricultural Sciences in Jiangsu , Yangzhou , China
Sistla Srinivas
Electronic publication date: 2023 Oct 4
Publication date: 2023
Volume: 11
Electronic Location ID: e16225
Received 2023 Jun 2; Accepted 2023 Sep 12
Copyright: © 2023 Han et al.
Copyright year: 2023
Copyright holder: Han et al.
License: This is an open access article distributed under the terms of the Creative Commons Attribution License, which permits unrestricted use, distribution, reproduction and adaptation in any medium and for any purpose provided that it is properly attributed. For attribution, the original author(s), title, publication source (PeerJ) and either DOI or URL of the article must be cited.
License URL: https://creativecommons.org/licenses/by/4.0/

Keywords: Hemolin, Metamorphosis, Immune recognition, Infection, Cnaphalocrocis medinalis

Funding: Jiangsu Agricultural Science and Technology Innovation Fund ZX(20)1004 Jiangsu Science Project of China BK20181215 Yangzhou science and technology project YZ2021034 This work was supported by the Jiangsu Agricultural Science and Technology Innovation Fund (Grant No. ZX(20)1004), the Jiangsu Science Project of China (Grant No. BK20181215) and the Yangzhou science and technology project (Grant No. YZ2021034). The funders had no role in study design, data collection and analysis, decision to publish, or preparation of the manuscript.

==============================
Background

As a member of the immunoglobulin superfamily, hemolins play a vital role in insect development and defense against pathogens. However, the innate immune response of hemolin to baculovirus infection varies among different insects.

Methods and results

In this study, the hemolin-like gene from a Crambidae insect, Cnaphalocrocis medinalis, CmHem was cloned, and its role in insect development and baculovirus infection was analyzed. A 1,528 bp contig as potential hemolin-like gene of C. medinalis was reassembled from the transcriptome. Further, the complete hemolin sequence of C. medinalis (CmHem) was cloned and sequenced. The cDNA of CmHem was 1,515 bp in length and encoded 408 amino acids. The deduced amino acid of CmHem has relatively low identities (41.9–62.3%) to various insect hemolins. However, it contains four Ig domains similarity to other insect hemolins. The expression level of CmHem was the highest in eggs, followed by pupae and adults, and maintained a low expression level at larval stage. The synthesized siRNAs were injected into mature larvae, and the CmHem transcription decreased by 51.7%. Moreover, the abdominal somites of larvae became straightened, could not pupate normally, and then died. Infection with a baculovirus, C. medinalis granulovirus (CnmeGV), the expression levels of CmHem in the midgut and fat body of C. medinalis significantly increased at 12 and 24 h, respectively, and then soon returned to normal levels.

Conclusions

Our results suggested that hemolin may be related to the metamorphosis of C. medinalis. Exposure to baculovirus induced the phased expression of hemolin gene in the midgut and fat body of C. medinalis, indicated that hemolin involved in the immune recognition of Crambidae insects to baculovirus.

Introduction

Hemolin, previously named P4 protein, belongs to the immunoglobulin (Ig) superfamily, which was described for the first time as cell adhesion molecule that can bind to the bacterial surface against bacterial challenge (Rasmuson & Boman, 1979; Sun et al., 1990). The expression of hemolin in insects induced by bacterial exposure has been extensively confirmed (Eleftherianos et al., 2007; Sun et al., 2015; Zuo et al., 2015; Liu et al., 2017; Orozco-Flores et al., 2017; Jung, Sajjadian & Kim, 2019; Admella & Torrents, 2022). Recently, it was proposed that hemolin also could be involved in viral defense. The mRNA level of hemolin was induced to be up-regulated after 24 h of ApNPV injection in Antheraea pernyi. Moreover, the antibacterial activity was not activated, suggesting that hemolin was involved in antiviral response as a virus inducible gene (Hirai et al., 2004). As-HEM and Ap-hemolin-like, homologous genes of hemolin, which were cloned from Bombycoidea insects, have also been shown to respond to baculovirus infection by increasing the mRNA expression level and protein production (Qian et al., 2017; Sun et al., 2015). Hemolin of Manduca sexta has also been confirmed to be involved in a polydnavirus infection, a virus attached to the injected eggs of parasitic wasps Cotesia congregate. A polydnavirus protein CcV1 can binds to hemolin and inhibits the hemolin function of binding lipopolysaccharide and agglutination of bacteria (Labropoulou et al., 2008). Of note, the hemolin expression of fat bodies and hemocytes in Helicoverpa zea and Helicoverpa virescens larvae infected with HzNPV was not significantly different from that of the control group (Terenius, Popham & Shelby, 2009). Hence, the expression of hemolin induced by viral infection varies among different insect families.

In addition to being induced by pathogens, hemolin has also been proven to be a necessary gene for development (Bettencourt, Terenius & Faye, 2002; Liu et al., 2017). Hemolin exists at all developmental stages of M. sexta. The content of hemolin in the hemolymph of naive larvae was the lowest and increased dramatically after pupation (Yu & Kanost, 1999). However, the relative mRNA levels of Spodoptera exigua hemolin were significantly different among various development stages, with the highest expression level in fourth instar larvae (Jung, Sajjadian & Kim, 2019). Silencing of hemolin gene in mated females of Hyalophora cecropia has resulted in deformed embryos that failed to hatch (Bettencourt, Terenius & Faye, 2002). In the pupa of Bombyx mori by injection of 4 μg/pupa dose siRNA of hemolin, 40% of the moths exhibited abnormal wing development (Liu et al., 2017). Moreover, as a hormone regulating insect development, 20-hydroxyecdysone (20E) can activate the expression of H. cecropia hemolin in the fat body of diapausing pupae, accompanied by ongoing protein synthesis (Roxstrom-Lindquist et al., 2005). These results indicated that the expression levels of hemolin are different in different insects and closely related to development.

Cnaphalocrocis medinalis granulovirus (CnmeGV) is an effective baculovirus agent against the rice leaffolder, a Crambidae insect, that it protects arthropods in field application (Xu et al., 2019). However, the progress of CnmeGV infection is slow, which limits its wide use. In this study, the hemolin gene (CmHem) of Cnaphalocrocis medinalis was cloned and characterized, its role in development and baculovirus infection was confirmed. This basic information is helpful to improve the understanding of hemolin and lay a foundation for further research on improving the infection efficiency of baculovirus by inhibiting hemolin.

Materials and Methods

Insect rearing and infection

The larvae of C. medinalis were fed with corn leaves in the laboratory, under standard conditions of 14:10 h (L:D) photoperiod, temperature 28 °C, at relative humidity of 70%. The eggs, different instars larvae, pupae and adults were collected. CnmeGV was purified and prepared for the different concentrations as previously described (Han et al., 2016). The early fourth-instar larvae were singled out and starved for 6 h, then fed with corn leaves soaked in 105 OB/ml CnmeGV (Han et al., 2021). Five infected larvae were collected as one replication after 12, 24, 48 and 96 h, respectively. Each treatment was replicated for three times. All larvae were dissected, and the midgut and fat bodies were washed in PBS prepared with diethyl pyrocarbonate treated H2O and collected, respectively. These tissues were immediately frozen in liquid nitrogen, then stored at −80 °C.

RNA extraction and reverse transcription

All frozen samples were pulverized in liquid nitrogen. Total RNA was extracted using RNA extraction kit (Code No. 9767) from TaKaRa. The purity and integrity of all RNA samples were assessed using a Nanophotometer® N50 Touch spectrophotometer (IMPLEN, Munich, Germany) and confirmed by 1% agarose gel electrophoresis, respectively. The cDNA of all samples was synthesized using the PrimeScript™ RT reagent kit with gDNA Eraser (Code No. RR047A) from TaKaRa (Shiga, Japan).

cDNA cloning

The hemolin-like unigenes were searched in the transcriptom database of C. medinalis using local BLAST of TB-tools software (Chen et al., 2020). The identified unigenes were reassembled by multiple alignment as original hemolin-like sequence of C. medinalis. A pair of primers were designed to amplify the full-length hemolin gene of C. medinalis (CmHem) (CmHemF: TGCCATTTTTGCTGTAGTTTTC; CmHemR: ATGAACCAGAGTTATGGGGATG), using the cDNA of fourth-instar larva as the template. PCR amplification was performed using the 2× Taq Master Mix polymerase (Code No. P112-02) from Vazyme on T100™ Thermal Cycler (Bio-rad, Hercules, CA, USA) with a condition of 95 °C for 30 s, following by 30 cycles of 5 s at 95 °C, 30 s at 50 °C, 60 s at 72 °C, extending 10 min at 72 °C. The PCR product was purified by 0.8% agarose electrophoresis, then cloned into pEASY-T3 vector and sequenced.

Reverse transcription quantitative PCR (RT-qPCR)

The relative expression level of CmHem (primers for DLHemF: GCCTTCAGAGGTGCTGTTCCG; DLHemR: TCGTCGTCTTTATGCCATTCGTA) was analysed quantitatively using comparative CT (2−ΔΔCT) (Livak & Schmittgen, 2001). The house keeping gene β-actin was used as an internal control for normalization (Han et al., 2021). RT-qPCR was performed using TB Green™ Fast qPCR Mix polymerase (Code No. RR430S) from TaKaRa on StepOnePlus Real-Time PCR System (Applied Biosystems, Waltham, MA, USA).

RNA interference (RNAi) of CmHem

Two pairs of specific dsRNA of CmHem, siRNA983 (sense: 5′-GGAGUAUAAGUUCAACGUUTT-3′; antisense: 5′-AACGUUGAACUUAUACUCCTT-3′) and siRNA1323 (sense: 5′-GCGAGAUAAUUUGUCGACATT-3′; antisense: 5′-UGUCGACAAAUUAUCUCGCTT-3′) and the non-specific siRNA-c (sense: 5′-UUCUCCGAACGUGUCACGUTT-3′; antisense: 5′-ACGUGACACGYYCGGAGAATT-3′) as negative control were synthesized in GenePharma. All synthesized siRNAs were modified with 2′-Ome and prepared in ddH2O with concentrations of 1 μg/μL, respectively. The siRNA983 and siRNA1323 were mixed by equal volume for treatment. A total of 200 nL of mixed siRNA and siRNA-c were injected into the fat bodies of mature larvae by Nanoject III (Norwalk, CA, USA), respectively. Each treatment was replicated three times with twelve insect samples.

Sequence analysis

Unigenes of hemolin-like were reassembled using Vector NTI. The cDNA and amino acid sequences were analysed by GENEDOC (https://genedoc.software.informer.com/2.7/). The conserved domains were analyzed by conserved domain database (CDD) (www.ncbi.nlm.nih.gov/Structure/cdd) (Lu et al., 2020). The predicted molecular weight and tertiary structure of CmHem protein were analysed by ExPASy (https://www.expasy.org/). The signal peptide was predicted using SignalP 5.0 server (Petersen et al., 2011). Multiple alignment was created by MUSCLE (Edgar, 2004) and the phylogeny of hemolin protein was built by maximum likelihood method using MEGA 7.0 (Kumar, Stecher & Tamura, 2016).

Results

Potential hemolin-like gene in the C. medinalis transcriptome

A total of 13 unigenes were identified in the transcriptome of C. medinalis by searching annotation files (Table 1). Most unigenes were short in length, ranging from 300 to 500 bp. The longest unigene was 676 bp, sharing 50.5% identity with the hemolin gene of Ostrinia furnacalis. The 13 unigenes were reassembled to obtain a new 1,528 bp contig as a potential hemolin-like gene of C. medinalis (Fig. 1). The newly assembled contig had 48 mutation sites. These undetermined bases need to be further confirmed by sanger dideoxy sequencing.

Table 1 The hemolin-like unigenes in the transcriptome of C. medinalis.

geneID	Length (bp)	Identity %	E-value	Score	NR_Description	
TRINITY_DN306890_c0_g1_i1	293	57.8	1.10E−20	107.8	hemolin (Ostrinia furnacalis)	
TRINITY_DN403619_c12_g2_i1	324	73.6	4.80E−41	175.6	hemolin (Ostrinia furnacalis)	
TRINITY_DN403619_c13_g2_i2	363	72	7.30E−46	191.8	hemolin (Ostrinia furnacalis)	
TRINITY_DN403619_c13_g5_i1	440	69.4	3.00E−54	219.9	hemolin (Ostrinia furnacalis)	
TRINITY_DN403619_c13_g13_i1	318	77.1	9.50E−42	177.9	hemolin (Ostrinia furnacalis)	
TRINITY_DN403619_c13_g14_i1	676	50.5	6.40E−19	103.2	hemolin (Ostrinia furnacalis)	
TRINITY_DN403619_c13_g14_i2	486	52.9	1.70E−18	101.3	hemolin (Ostrinia furnacalis)	
TRINITY_DN403619_c13_g40_i1	316	46.8	8.10E−17	95.1	hemolin (Danaus plexippus)	
TRINITY_DN403619_c13_g46_i1	286	51.6	8.30E−21	108.2	hemolin (Ostrinia furnacalis)	
TRINITY_DN403619_c13_g53_i1	447	50	5.10E−33	149.4	hemolin (Ostrinia furnacalis)	
TRINITY_DN403619_c13_g69_i1	310	65.3	3.50E−33	149.4	hemolin (Ostrinia furnacalis)	
TRINITY_DN487348_c0_g1_i1	302	56.4	5.70E−04	52.4	hemolin (Samia ricini)	
TRINITY_DN240507_c0_g1_i1	216	72.1	7.20E−17	94.7	hemolin (Ostrinia furnacalis)	

Figure 1 The reassembly (A) and cloning (B) of hemolin-like gene in C. medinalis transcriptome.

The 1,515 bp sequence of CmHem has 1,227 bp of 408 amino acid protein-coding region, 53 bp putative 5′ untranslated region and 235 bp 3′ untranslated region. The signal sequence is blue circle. Four groups of the immunoglubin domain (Ig1, 2, 3, 4) are green boxes.

Cloning and characterization of the hemolin gene from C. medinalis

The hemolin gene of C. medinalis (CmHem) was amplified by primers designed with the assembled conting as the reference sequence. The cloned CmHem gene was 1,515 bp in length and encoded 408 amino acids (GeneBank accession number: MK138364). CmHem protein was predicted to contain a signal peptide of eight amino acids (QAQPVSQA). The predicted isoelectric point and molecular weight were 5.77 and 45.17 kDa, respectively. CmHem has four Ig domains (IG1, IG2, IG3 and IG4) by CDD analysis (Fig. 1).

Multiple alignment results showed that the Hemolin protein from C. medinalis shared the most similarity with that from O. furnacalis (62.3% identity) and the lowest similarity with that from Plutella xylostella (41.9% identity). Similar to other hemolin proteins, CmHem contains eight cysteine residues forming four disulfide bridges with a tryptophan residue packed against the disulfide bond (Fig. 2A). The tertiary structure prediction showed that CmHem had 47.1% identity with the Hemolin of H. cecropia, both of which had four Ig domains (Fig. 2B). A total of 22 related hemolin protein sequences were downloaded from GeneBank for evolutionary analysis. Phylogenetic analysis indicated that the CmHem had the close genetic distance with the hemolin of O. furnacalis. In addition, different from Lepidoptera, L. vannamei belonging to the crustacean, was grouped into a separate cluster (Fig. 2C).

Figure 2 The characteristic of CmHem protein from C. medinalis.

(A) Multiple alignment of hemolin proteins between C. medinalis and other insects. Pentagrams and triangles point to cysteine residues and tryptophan residues, respectively. (B) Predicted tertiary structure of CmHem protein by ExPASy. The CmHem protein contains four immunoglobulin domains (IG1, IG2, IG3 and IG4). (C) Phylogenetic analysis of hemolin proteins from C. medinalis and other species. The phylogenetic tree was constructed by MEGA 7.0 using maximum likelihood method.

Expression levels of CmHem at different developmental stages

To clarify the effect of hemolin gene on insect development, the expression levels of CmHem in C. medinalis at different development stages were analysed by RT-qPCR. Using β-actin as reference gene, the relative expression level of CmHem in fifth instar larvae is set as 1. The results showed that the expression level of CmHem was higher in eggs, pupae and adults, while lower in larvae. In particular, the expression level of CmHem in eggs was 310 times higher than that of fifth instar larvae and three times higher than that of pupae or adults. However, it was no significant difference in the CmHem expression among different larval stages (Fig. 3).

Figure 3 The relative expression levels of CmHem at different developmental stages of C. medinalis.

1st, 2nd, 3rd, 4th and 5th represent different instar larvae. The values are represented as mean ± SD. Using β-actin as reference gene, the relative expression level of CmHem in fifth instar larvae is set as 1. Different small letters above histograms indicated significant differences among different treatments at the 0.05 level (ANOVA).

The RNAi of CmHem in mature larvae of C. medinalis

Using injection of siRNA-c as control group, after 48 h of CmHem siRNAs injection, the abdominal somites of larvae became straightened and could not pupate normally, then died on day 3 onwards (Fig. 4A). Further, the relative expression of CmHem was analysis after 48 h of RNAi. RT-qPCR was used to determine the expression level of hemolin relative to β-actin. Compared to the control group, the CmHem transcription of the RNAi group decreased by 51.7% (Fig. 4B).

Figure 4 The phenotype of mature larvae after CmHem interference at 72 h (A) and the relative expression levels of CmHem after 48 h of interference (B).

The CK is the control group injected with siRNA-c. The hemolin RNAi is the treatment group injected with siRNA983 and siRNA1323 of CmHem. RT-qPCR was used to determine the expression level of hemolin relative to β-actin. Asterisks indicated difference statistically significant at 0.01 level (student’s t-test).

Expression analysis of CmHem in C. medinalis after infection with CnmeGV

The immune response of CmHem to CnmeGV challenge in the fat body and midgut tissues was analyzed at different stages of infection. The expression level of CmHem in midgut tissue of C. medinalis was up-regulated after 12 h of infection, with 22 times higher than that before infection. Subsequently, the expression of CmHem decreased without significant difference from that before infection. However, in fat body, the expression level of CmHem was up-regulated four times higher at the time after 24 h of infection. After that, the expression level of CmHem in fat body also returned to the level before infection (Fig. 5).

Figure 5 The relative expression levels of CmHem in midguts (A) and fat bodies (B) of C. medinalis after CnmeGV infection.

Samples were collected after 12, 24, 48 and 96 h of infection for differential expression analysis of CmHem (ANOVA, p < 0.0001). The β-actin was used as reference gene. RT-qPCR was used to determine the expression level of hemolin relative to β-actin. Different small letters above histograms indicated significant differences among different treatments at the 0.05 level (ANOVA).

Discussion

Since hemolin was first characterized in H. cecropia, more and more homologs of hemolin have been cloned from other insects (Sun et al., 1990). In this study, a homologous gene of hemolin from C. medinalis (CmHem) was identified. CmHem is composed of four Ig domains to form a horseshoe structure, which is consistent with other insect hemolins (Qian et al., 2017; Sun et al., 2015). However, it is different from the LvHemolin of L. vannamei, which is composed of seven Ig domains. The last four Ig domains of LvHemolin share high identity with insect hemolins and form a globular structure, while the role of other three Ig-domains is still unknown (Zuo et al., 2015). As a member of the immunoglobulin superfamily, Down’s syndrome cell adhesion molecule (DSCAM) also contains 10 Ig domains, and is speculated to be a hypervariable pattern-recognition receptor in insect immunity (Ng & Kurtz, 2020; Watson et al., 2005). The external facing Ig2 and Ig3 of horseshoe structure of DSCAM play an important role in heterophilic specific binding with pathogens (Li et al., 2018; Ng & Kurtz, 2020). Hence, the Ig2 and Ig3 domains of CmHem may also be involved in pathogen recognition.

Hemolin was expressed in all stages of insects development (Yu & Kanost, 1999; Jung, Sajjadian & Kim, 2019). In our study, the expression of CmHem in eggs, pupae and adults of C. medinalis was significantly higher than that of larvae. Similarly, the synthesization of M. sexta hemolin was very low during the larval feeding periods, but very high in eggs and pupae (Yu & Kanost, 1999). However, the relative mRNA level of hemolin in eggs of S. exigua was found to be the lowest (Jung, Sajjadian & Kim, 2019). Although its expression levels were significantly different among various developmental stages, any clear physiological significance is unknown about these developmentally-induced hemolins. Previous studies have found that the expression of hemolin in H. cecropia can be induced by 20E (Roxstrom-Lindquist et al., 2005) and involved in intercelluar adhesion, including cell proliferation and wound healing (Sato et al., 2016), indicating that hemolin is related to insect development. In insect metamorphosis, specialized structures of the preceding developmental stage have to be completely remodeled to accommodate new structures and behaviors (Pinet & McLaughlin, 2019). We suppressed the CmHem expression by RNAi, and the mature larvae could not pupate normally. Further, Plodia interpunctella hemolin is expressed only in epidermis, suggesting its functional association with metamorphosis (Aye et al., 2008; Shafeeq, Ulabdin & Lee, 2017). The proteomic analysis of hemolymph proteins during larva-to-pupal metamorphosis of B. mori further supports the functional role of hemolin in the metamorphosis process, as many immune-associated proteins, including hemolin, were found to be related to metamorphosis (Hou et al., 2010). These results indicated that hemolin may play a key role in the metamorphosis of insects.

Hemolin can bind to lipopolysaccharide (LPS) or lipoteichoic acid on the bacterial surface, and aggregate them to lead hemocytes reaction (Daffre & Faye, 1997; Sun et al., 1990). Escherichia coli, Beauveria bassiana (Sun et al., 2015), Micrococcus luteus (Qian et al., 2017), Photorhabdus temperata (Jung, Sajjadian & Kim, 2019) have been widely proved to induce the expression of insect hemolin. Interestingly, hemolin can respond to the viral infection in the superfamily Bombycoidea, but not in the Noctuidae (Qian et al., 2017; Terenius, Popham & Shelby, 2009). Our results showed that CmHem in the midgut and fat body of C. medinalis can responded to CnmeGV infection. However, the expression of CmHem soon returned to normal levels after up-regulation, indicating that CmHem may involved in the initial immune recognition. Whether LPS analogs are present on the surface of baculovirus is not clear, or hemolin can bind to unknown proteins on the surface of the virus. The study found that hemolin isolated from H. cecropia, was confirmed as a specific lectin by the homophilic binding properties analysis and bound to glycosylated surfaces, such as the virion envelope (Bettencourt et al., 1999).

As a pattern recognition receptor, hemolin is more likely to regulate and stimulate the humoral and cellular immune response through activating a series of signal pathways (Terenius, 2008). In Drosophila cell line mbn-2, hemolin enhanced the protein kinase C (PKC) activity in hemocyte crude extracts and prevented tyrosine phosphorylation of two proteins of 35 and 40 kDa, suggesting that hemolin is involved in the regulation of the cellular immune responses via a pathway that includes PKC activation and protein tyrosine phosphorylation (Lanz-Mendoza et al., 1996). The melanization reaction is a principal humoral immune response in insects (Eleftherianos et al., 2021). In A. pernyi, knockdown of hemolin regulated the expression level of antimicrobial peptide genes and decreased prophenoloxidase activation in hemolymph stimulated by microbial invaders. (He et al., 2022). Hemolin in the pupae of H. cecropia has also been shown to be important for the triggering of the prophenoloxidase cascade in the defence against bacterial infections (Terenius et al., 2007). In addition, The third intron of H. cecropia hemolin contains κB motif, which can be activated by Relish of Toll signalling pathway (Roxström-Lindquist et al., 2002). Inhibition of Toll pathway in S. exigua also reduced the expression of hemolin (Park & Kim, 2012). These results indicated that the expression of hemolin is related to Toll signaling pathway. There are 10 unigenes in Toll/IMD pathway, two unigenes in JNK pathway, eight unigenes in JAK/STAT pathway were identified in C. medinalis infection with CnmeGV (Han et al., 2021). Toll, IMD and JAK-STAT pathways have been shown to be associated with viral infection (Jakubowska, Vogel & Herrero, 2013; Liu et al., 2015). Therefor, the study of the expression of hemolin and the interaction of these signalling pathways is worth exploring.

Conclusions

In conclusion, we have cloned and partially characterized the full length cDNA of hemolin from C. medinalis. Hemolin plays an important role in metamorphosis of C. medinalis. Baculovirus CnmeGV infection induced the phased expression of hemolin gene in midgut and fat body of C. medinalis.

Supplemental Information

Supplemental Information 1 Raw data.

Click here for additional data file.

The authors are grateful to Yurong Lu and Yangsheng Cai for their assistance in laboratory work.

Additional Information and Declarations

Competing Interests

Author Contributions

DNA Deposition

Data Availability

The authors declare that they have no competing interests.

Guangjie Han conceived and designed the experiments, performed the experiments, analyzed the data, prepared figures and/or tables, authored or reviewed drafts of the article, and approved the final draft.

Chuanming Li performed the experiments, prepared figures and/or tables, and approved the final draft.

Nan Zhang performed the experiments, prepared figures and/or tables, and approved the final draft.

Qin Liu performed the experiments, authored or reviewed drafts of the article, and approved the final draft.

Lixin Huang performed the experiments, prepared figures and/or tables, and approved the final draft.

Yang Xia performed the experiments, prepared figures and/or tables, and approved the final draft.

Jian Xu conceived and designed the experiments, prepared figures and/or tables, authored or reviewed drafts of the article, and approved the final draft.

The following information was supplied regarding the deposition of DNA sequences:

The sequences are available at GenBank: MK138364.

The following information was supplied regarding data availability:

The data is available at NCBI: PRJNA555407.

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
