# Peer review of "CmHem, a hemolin-like gene identified from Cnaphalocrocis medinalis, involved in metamorphosis and baculovirus infection"

_PeerJ, doi:10.7717/peerj.16225_

## Round 0.1 · original submission · Major Revisions

Please follow the instructions from reviewers and resubmit at the earliest.

Reviewer 1 ·

Basic reporting

The paper studied the function of Hemolin, a member of the immunoglobulin (Ig) superfamily, in the development of insects and their response to baculovirus infection. This study focuses on the hemolin-like gene (CmHem) found in Cnaphalocrocis medinalis, a Crambidae insect. The authors successfully cloned and sequenced the CmHem gene, which encoded a protein of 408 amino acids. Despite relatively low sequence identities to other insect Hemolins, CmHem shared the characteristic four predicted Ig domains found in Hemolins. Expression analysis revealed that CmHem exhibited differential expression patterns throughout different developmental stages, with the highest expression in eggs. Moreover, RNA interference experiments demonstrated the importance of CmHem in the metamorphosis of C. medinalis, as interference led to abnormal pupation and subsequent death. Furthermore, the study investigated the response of CmHem to baculovirus infection. The expression levels of CmHem in the midgut and fat body of C. medinalis significantly increased at different time points after infection, suggesting its involvement in the immune recognition of Crambidae insects to baculovirus. The expression returned to normal levels thereafter, indicating a phased response. The findings contribute to our understanding of the multifaceted roles of Hemolin in Cnaphalocrocis medinalis development and immune responses.

Experimental design

The paper encompasses four experimental procedures: the PCR-based cloning of the CmHem gene, RT-PCR analysis of CmHem gene expression at various developmental stages of C. medinalis, observation of C. medinalis development following interference of CmHem through RNAi, and analysis of CmHem expression after C. medinalis granulovirus (CnmeGV) infection. The results of the paper contribute to people’s understanding of CmHem gene expression in C. medinalis. However, the study's contribution to understanding the role of CmHem in C. medinalis development and immunity against insects remains limited. Specifically, there is no characterization of the in vivo functionality of Ig discovered in CmHem, and the molecular-level mechanisms by which CmHem functions against viruses are not elucidated.

Validity of the findings

The writing style employed in the paper raises significant concerns. For example, in the background section of the abstract, the author states, "Hemolins, as members of the immunoglobulin (Ig) superfamily, play a vital role in insect development and defense against pathogens. However, Hemolin has different effects on insect development and the innate immune response to baculovirus infection." This description lacks coherence and fails to establish a clear connection between the two scenarios. Similarly, the first and third paragraphs of the introduction suffer from the same disjointed writing style and require a complete overhaul.

Moreover, while the discussion section provides additional background information on current Hemolin research progress, it fails to establish a clear connection to the specific study presented in this paper. The writing in this section also appears cumbersome and needs improvement. I recommend that the author of the paper seeks the assistance of a fluent English speaker to review and revise the paper before its final submission. This will help ensure that the writing is clear, coherent, and free from any language-related issues that may hinder the readers' understanding and engagement with the content.

Annotated reviews are not available for download in order to protect the identity of reviewers who chose to remain anonymous.

Reviewer 2 ·

Basic reporting

In this manuscript, the authors identified a hemolin-like gene CmHem from Cnaphalocrocis medinalis and showed it was involved in metamorphosis and baculovirus infection. The manuscript is well-written in English, and the background information and references are adequately introduced and cited. Overall, the manuscript is highly intriguing and the findings hold great value for the field. I only have a few specific comments.

1. L53-L54: “However, its role in response to baculovirus infection is still controversial.”
Why it is important for baculovirus infection and list the brief controversial results and references.
2. L62 “....for development” please list more details for which kind of development?
3. L72-79: It’s not very close to this study aims, should shorten these sentences or delete them.
4. L123: 200 nL? It’s correct? How could you inject 200 nL?
5. L133-144: merge Fig. 1 and 2 into a figure. And need not show the sequence of CmHem gene, use a carton figure similar to Fig. 1 to show the different parts of Fig. 2.
6:. Fig. 4 and 5 with “a or b” is not the normal style to express significant differences (*, **.....) or no significant differences (ns). The figure legends of Fig. 5 and 6 were too simple, need to give more details to make it easy to understand by readers.
7. L153-158: It’s not clear. Need to add more details to this part to show how mRNA was detected? And which development mRNA expression was used as a comparison group?
About Fig. 4: relative expression level (Y axis) is really low, which control group you compared to? Please give brief information in the figure legends.
8. L159: give the full name of CK.
9. L162: You can’t conclude that “the RNAi efficiency of CmHem was 51.7%” only based on the CmHem transcription. Rewrite this sentence “CmHem transcription of RNAi group decreased (or showed...decrease) ......compared to...”
10. L163-169: Forgot to label Fig. 6 in this part of the text.
11: L223-233: need to rewrite conclusions to make it more clear and concentered on your findings in this study.

Experimental design

As list 1

Validity of the findings

As list 1

Additional comments

As list 1

---

## Round 0.2 · accepted · Accept

All comments were revised satisfactorily and I recommend accept.

Reviewer 1 ·

Basic reporting

The paper studied the function of Hemolin, a member of the immunoglobulin (Ig) superfamily, in the development of insects and their response to baculovirus infection. This study focuses on the hemolin-like gene (CmHem) found in Cnaphalocrocis medinalis, a Crambidae insect. The authors successfully cloned and sequenced the CmHem gene, which encoded a protein of 408 amino acids. Despite relatively low sequence identities to other insect Hemolins, CmHem shared the characteristic four predicted Ig domains found in Hemolins. Expression analysis revealed that CmHem exhibited differential expression patterns throughout different developmental stages, with the highest expression in eggs. Moreover, RNA interference experiments demonstrated the importance of CmHem in the metamorphosis of C. medinalis, as interference led to abnormal pupation and subsequent death. Furthermore, the study investigated the response of CmHem to baculovirus infection. The expression levels of CmHem in the midgut and fat body of C. medinalis significantly increased at different time points after infection, suggesting its involvement in the immune recognition of Crambidae insects to baculovirus. The expression returned to normal levels thereafter, indicating a phased response. The findings contribute to our understanding of the expression of Hemolin in Cnaphalocrocis medinalis development and immune responses.

Experimental design

The paper encompasses four experimental procedures: the PCR-based cloning of the CmHem gene, RT-PCR analysis of CmHem gene expression at various developmental stages of C. medinalis, observation of C. medinalis development following interference of CmHem through RNAi, and analysis of CmHem expression after C. medinalis granulovirus (CnmeGV) infection.

Validity of the findings

The validity of the findings in this paper is substantiated by the utilization of four distinct and well-established experimental procedures as described above. These methods served to validate the authors' objective of cloning CmHem and exploring its expression at various developmental stages and in response to infection. Additionally, they assessed the impact of CmHem interference in Cnaphalocrocis medinalis, providing clear evidence of its significant role in immune recognition.

Additional comments

The authors have revised the introduction and discussion sections of the paper, leading to a more coherent and reader-friendly presentation. Some typos have been highlighted in the attached document; please make the necessary revisions, especially paying attention to singular and plural forms.

Annotated reviews are not available for download in order to protect the identity of reviewers who chose to remain anonymous.

Reviewer 2 ·

Basic reporting

The authors have addressed my comments well. I have no more questions.

Experimental design

The authors have addressed my comments well. I have no more questions.

Validity of the findings

The authors have addressed my comments well. I have no more questions.